# Direct imaging of the circular chromosome in a live bacterium

Fabai Wu[1,2,3], Aleksandre Japaridze[1,3], Xuan Zheng[1], Jakub Wiktor[1], Jacob W. J. Kerssemakers[1] & Cees Dekker [1]

Although the physical properties of chromosomes, including their morphology, mechanics, and dynamics are crucial for their biological function, many basic questions remain unresolved. Here we directly image the circular chromosome in live *E. coli* with a broadened cell shape. We find that it exhibits a torus topology with, on average, a lower-density origin of replication and an ultrathin flexible string of DNA at the terminus of replication. At the single-cell level, the torus is strikingly heterogeneous, with blob-like Mbp-size domains that undergo major dynamic rearrangements, splitting and merging at a minute timescale. Our data show a domain organization underlying the chromosome structure of *E. coli*, where MatP proteins induce site-specific persistent domain boundaries at Ori/Ter, while transcription regulators HU and Fis induce weaker transient domain boundaries throughout the genome. These findings provide an architectural basis for the understanding of the dynamic spatial organization of bacterial genomes in live cells.

[1] Department of Bionanoscience, Kavli Institute of Nanoscience Delft, Delft University of Technology, Van der Maasweg 9, 2629 HZ Delft, The Netherlands. [2] Division of Geological and Planetary Sciences, California Institute of Technology, 1200 E California Blvd, Pasadena, CA 91125, USA. [3] These authors contributed equally: Fabai Wu, Aleksandre Japaridze. Correspondence and requests for materials should be addressed to C.D. (email: c.dekker@tudelft.nl)

I t is increasingly understood that the spatial organization of a genome is imperative for its biological function. Statistical analyses of fluorescence foci localization[1–4] and Hi-C contact frequencies[5–9] of genomic loci in ensembles of cells have yielded great progress towards understanding the organizational principles of bacterial chromosomes. Yet, even for the well-studied model bacterium *Escherichia coli*, many basic questions remain unresolved regarding the chromosomal (sub-)structure[2,10], its mechanics and dynamics[11,12], and the link between structure and function. Various recent experiments have led to different proposals for the structure of the highly compact 4.6-Mbp circular genome of *E. coli* (Fig. 1a). For example, while the macrodomain model suggested large (~0.5–1Mbp) domains induced by long-range interactions[10,13,14], the linear filament model depicted a rather uniformly stacked nucleoid body connected by a thin terminal string[2,15,16].

Whole-chromosome imaging[11,17,18] would be an ideal tool to resolve and characterize the fine structure of chromosomes. Unfortunately, such studies suffer from a limited resolution because the chromosome is tightly confined within the rod-shape cell that is narrower than 1 μm, and furthermore the common modus of high-resolution imaging of fixed cells[19–21] prevents capturing the internal dynamics of the chromosome that is continuously replicating upon cell growth. We set out to overcome these limitations in two ways: First, we used the MreB-protein-inhibitor drug A22 to disrupt the formation of new cytoskeleton for the cell, which led to cells that grew wider in size, while staying alive in a fully physiologically active state[22] (Supplementary Fig. 1). Next, by using a strain with a *dnaC2(ts)* allel[23], we stopped initiating DNA replication at 40 °C by preventing the loading of the DnaB helicase onto the origin of replication[24], and therefore cells could not initiate a new round of replication but merely finished already initiated rounds. As a result, the vast majority (>80%) of cells maintained only one single chromosome while growing from a rod into a lemon shape (~2-μm wide, ~4-μm long, and ~1-μm high under an agarose pad) over the course of 2–3 h (Fig. 1b).

## Results

### Visualization of the circular chromosome by cell widening.
Interestingly, upon a two-fold widening of the cell, the single *E. coli* chromosome was observed to laterally expand and gradually open up into a torus (Fig. 1b). This topology was consistently observed through different imaging techniques such as wide-field epifluorescence and (2D and 3D) Structured Illumination Microscopy (SIM) (Fig. 1c, d, Supplementary Figs. 2 and 3), and with different fluorescent labels in live cells (Fig. 1e). These images of an open ring-like geometry confirmed that two chromosome arms flanking the origin of replication in *E. coli* are not cross-linked[15], an arrangement distinct from the SMC-mediated arm zipping that was reported for *Caulobacter crecentus*[25] and *Bacillus subtilis*[18]. Note that the toroidal geometry is not trivial, since a priori, cell widening could have been expected to lead to a homogeneously spread-out globular cloud of DNA[26], an unaltered ellipsoid[2], or a stiff arc[11]. Moreover, the chromosomes were able to immediately resume replication and cell division after a brief reactivation of the DnaC protein (where after 10 min re-incubation at room temperature ~60% of all cells initiated replication, Supplementary Fig. 6), during which the replicated regions branched out while conserving their bundle morphology (Fig. 1f). By contrast, upon treatment for a short time with drugs such as rifampicin (which blocks transcription by inhibiting RNA Polymerase) or ciprofloxacin (which impedes the homeostasis of supercoiling through inhibiting TopoIV and gyrase activity), or upon induction of the stationary phase, the

chromosomes collapsed and generally lost the torus topology (Supplementary Fig. 6A). The torus topology was not dependent on the slightly elevated temperature (40 °C) used to maintain a single chromosome in the (dnaCts) cells (as evidenced by experiments performed at 30 °C with CRISPR-inhibited replication initiation; Supplementary Fig. 6B), nor was it unique to the AB1157 strain used in the experiments (as it also appeared in the MG1655 strain; Supplementary Fig. 7A). Importantly, once the A22 drug was removed and the cells were transferred to 30 °C, they could regain growth and recover their rod shape, indicating that the cells were fully alive during the imaging (Supplementary Fig. 1). All of this indicates that it is a general feature for a single chromosome in widened *E. coli* cells. We conclude that the torus topology is maintained by active physiological processes, and hence serves as an excellent model object for resolving the organizational principles of a *E. coli* chromosome in live cells.

The direct visualization of the genome allowed us to quantitatively measure the width and length of the *E. coli* chromosome bundle (Fig. 1g, i). Facilitated by deconvolution which reduced the out-of-focus background intensity in wide-field imaging (Fig. 1c, Supplementary Fig. 3), we mapped the ridge line (Supplementary Fig. 4) of the chromosome, and measured the length along this contour. The average chromosome contour length was found to be $4.0 \pm 0.6$ μm (mean ± s.d., Fig. 1h, Supplementary Fig. 3, $n = 269$), while the average bundle thickness, characterized by the average full-width-at-half-maximum along the chromosome (FWHM), was $0.45 \pm 0.05$ μm (Fig. 1i). The chromosome contour length as measured by 2D-SIM yielded similar values (Supplementary Fig. 5). These data provide useful input for future modeling of the polymer structure of the chromosome under weak confinement and volume exclusion in the segregation of newly replicated DNA[2,11,27].

### *E. coli*'s chromosome is highly heterogeneous in structure.
The donut-shape chromosomes were observed to be strikingly non-uniform. The DNA density was heterogeneous along the circumference, partitioned into blob-like domain structures (Fig. 2a). Using a custom-made cluster-analysis script (, Supplementary Figs. 8 and 9), we found that each single chromosome contains between 3 and 8 apparent domains (Supplementary Fig. 10), with 4 as the most probable number in a distribution described by a lognormal probability density function (PDF) (Fig. 2b). The blobs showed a broad range of physical sizes, with a diameter D ranging from 200 nm to 1 μm (mean ± s.d. = $0.6 \pm 0.20$ μm, Fig. 2c). Next, we quantified the DNA length L (measured in base pairs) contained in each cluster based on fluorescence intensity, yielding values from 150 kbp to above 3 Mbp (Fig. 2d). Up to DNA length scales of more than 1 Mb, D was found to scale with L according to a $D \sim L^\alpha$ power law with an exponent $\alpha = 0.60 \pm 0.04$ (Fig. 2e), a scaling property that is surprisingly similar to that of a self-avoiding polymer[28].

In order to quantify the average DNA density as a function of the genomic sequence coordinate, we mapped the HUmYpet fluorescence intensity along the ridges of the donut-shape chromosomes (Supplementary Figs. 11–13), aided by fluorescence repressor operator system (FROS) markers (Figs. 1a and 3a). Note that HU binds uniformly to the chromosome at the ~200 nm scale of our resolution (i.e., slight preferences for AT-rich sequences[29] at the nm scale can be ignored), and hence the fluorescence intensity is an excellent estimate for the local DNA density (Supplementary Fig. 11). Figure 3a–c shows the data from a strain with labels at the L3 and R3 positions[15], which divide the circular chromosome into an *oriC*- and *dif*-containing branch that have an intensity ratio of 72:28% ($n = 84$, standard error 1%), close to the expected 70:30% DNA ratio. Next, we mapped

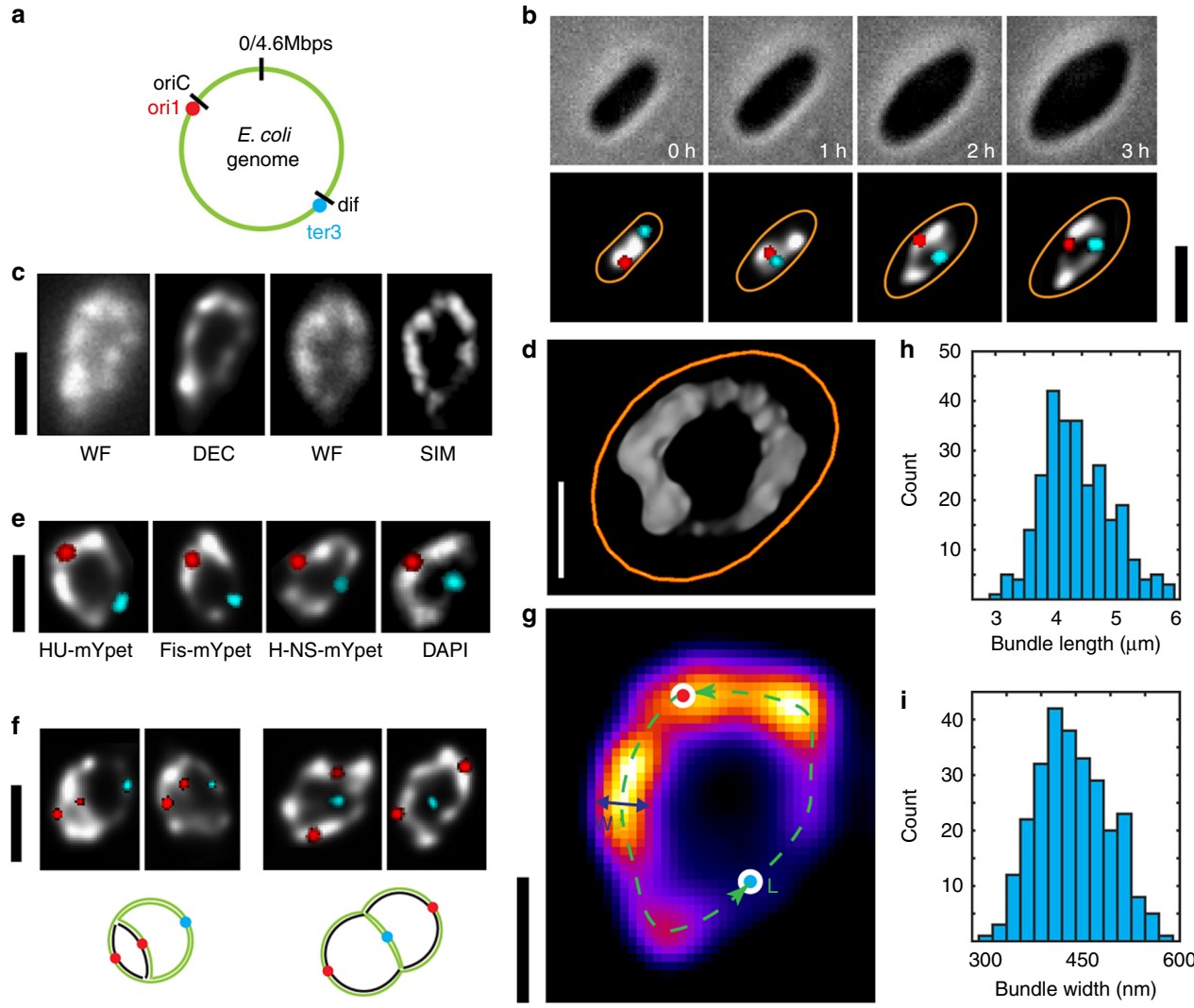

**Fig. 1** The circular *E. coli* chromosome exhibits a toroidal donut-shape that can be visualized upon cell expansion. **a** Schematic of a *E. coli* genome. Two FROS markers are shown in red (Ori1 marked by lacO arrays, targed by LacI-mCherry) and cyan (Ter3 marked by tetO arrays, targeted by TetR-mCerulean). **b** Time-lapse fluorescence images of an *E. coli* (*dnaC2(ts)* allel) cell growing into a lemon shape at 40 °C under A22 treatment. Top panel, phase contrast image; bottom panel, overlay of Ori focus (red) and Ter focus (cyan) on a grey-scale deconvolved image of the chromosome labeled by HU-mYPet. Time is indicated in hours. **c** Fluorescence images showing two opened circular chromosomes captured by different methods. Left: WF, wide-field image, and DEC, deconvolved image of that WF. Right: WF and SIM, structured-illumination microscopy image of that WF image. **d** Donut-shape chromosome of *E coli*, as imaged in 3D-reconstructed SIM. Orange outlines the cell contour. **e** Similar donut-shape chromosome images are obtained for different DNA-binding fluorescent labels (HU-mYPet, Fis-mYPet, H-NS-mYPet, DAPI; all DEC images). **f** Fluorescence images of early (left) and late (right) stages of DNA replication of circular chromosomes. Bottom: cartoon illustrations; black strands indicate newly replicated DNA. **g** Fluorescent image of a donut-shape *E. coli* genome shown as a heat map. Indicated are the ridge of the bundle (green dashed line), the oriC and dif genomic loci near the origin and terminus of replication (red and blue dots respectively), and the bundle width (blue line). **h** Histogram of chromosome bundle lengths measured along the bundle ridge (cf. panel G). $n = 269$. **i** Histogram of the average chromosome bundle widths quantified as the full-width-at-half-maximum of the peak intensity across the donut chromosomes. $n = 269$. Scale bars in B/C/E/F, 2 μm. Scale bars in D/G, 1 μm

the contour coordinates onto the genome sequence by constructing a cell-average cumulative density function that starts and ends at the L3 foci, which allowed physical positioning of the genomic loci including *oriC* and *dif* sites (where DNA replication initiates and terminates, respectively) onto the torus (Fig. 3d).

This yielded the average DNA density profile, which displays a pronounced M-shape curve with a very deep minimum located at the *dif* locus and a second, less deep yet well developed, minimum at the *oriC* locus (Fig. 3e). Interestingly, at the global density minimum which consistently resides near the *dif* site (Fig. 3c, d), a mere 2% of the genome (92kbp) spans as much as 11 ± 6% of the physical bundle length (Fig. 3e, inset). These *oriC* and *dif* loci

are connected by the DNA-dense left and right arms, which show a slight asymmetry with a somewhat higher DNA density peak in the left arm. The global M-shape as well as the same locations of the minima were also found in a second independent strain with Ori1 and Ter3 labels located adjacent to the *oriC* and *dif* sites, albeit with less well distinguishable left/right arms due to symmetry (Fig. 3f, Supplementary Fig. 11). The average M-shape was also conserved in experiments with multiple different fluorescent labels (Supplementary Fig. 14) indicating that the low-density regions were not due to the chromosome labelling used, neither to the positioning of tetO and lacO repeats (Figs. 1b, 3e, f, Supplementary Fig. 15).

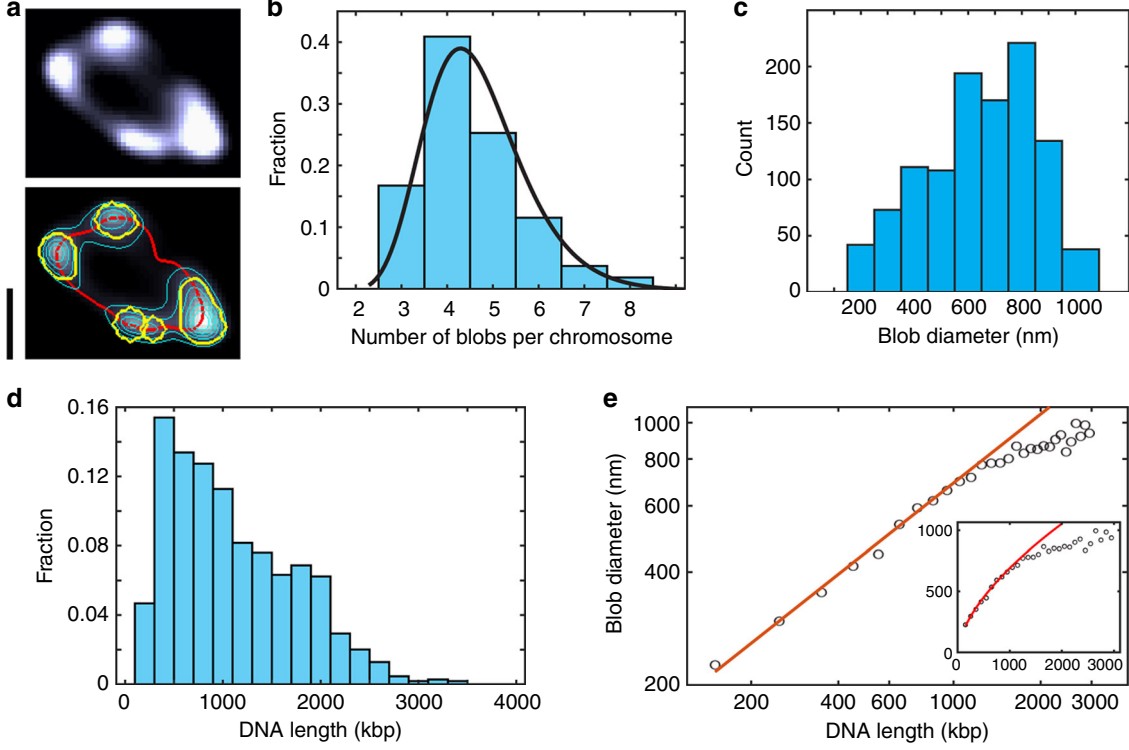

**Fig. 2** Domain distribution within the circular chromosome of *E. coli*. **a** Example of the automated domain recognition. Top panel, fluorescent image of a toroidal chromosome (deconvolved image). Bottom panel, same image with ridge line (red), equal-intensity lines (thin cyan lines), and blob boundaries (yellow). Scale bar, 2 μm. **b** Number of blobs per chromosome. Black line shows the probability density function (PDF) of a fitted lognormal distribution. $n = 269$. **c** Bar plot of the PDF of the physical size (diameter) of the blobs. **d** Distribution of domain sizes (DNA length in kbp) in all measured cells. **e** Blob diameter $D$ (nm) plotted versus the DNA length $L$ (kbp) contained in each blob, on a log-log scale. Circles indicate mean calculated for all cells within a 100 kbp bin size. Red line is a fit of a power law, $D \sim L^\alpha$ with $\alpha = 0.60 \pm 0.04$. Inset shows the same data plotted on linear scales

Individual DNA density plots of single chromosomes (cf. Fig. 3c) exhibited a high variety of local maxima and minima, from which we extracted the centre positions of the genomic locations of regions with high DNA density ('domains') and regions with low DNA density ('domain boundaries'), respectively. It is reasonable to denote these regions as 'domains' because they correspond to the blob-like domain structures that are visible by microscopy. As shown in Fig. 3g, very pronounced domain boundaries were consistently found at the *oriC*- and *dif*-proximal regions, whereas the prominent domain centres were found near the centres of the left/right arms. By contrast, less pronounced domain centres and boundaries were found to distribute more evenly throughout the genome (Fig. 3h). This led us to hypothesize that different mechanisms may be at play in defining the chromosomal domain structure in *E. coli*: (i) a mechanism that reduces DNA condensation at the *oriC* and *dif* regions or promotes interactions at the centres of the two arms, and (ii) a mechanism causing transient dynamic domains across the genome—both of which we discuss below.

**MatP protein induces domain boundaries at ori and ter**. We first explored the origin of the pronounced DNA density minima at the *dif* and *OriC* sites (Fig. 3e, g). As MatP proteins were recently implicated in mediating the actions of MukBEF SMC proteins and topoisomerase IV at the Ter and Ori regions[8,30], we examined the chromosome density distributions in Δ*matP* cells. Whereas broadened cells without MatP also exhibited toroidal chromosomes (Fig. 3j), we found that, strikingly, all distinct density peaks and gaps disappeared in the average density distribution (Fig. 3i, Supplementary Fig. 16) and instead the density

along the chromosome was uniform (Fig. 3i, j). Interestingly, local density peaks and gaps were still observed in individual cells, but they were evenly distributed across the chromosomes with no prominent features in either the *oriC* or *dif* sites (Fig. 3g, h, Supplementary Fig. 16). Notably, the thin terminal string persisted at different stages of the replication cycle (Fig. 1f) in wild-type, but not in Δ*matP* cells (Supplementary Fig. 16). MatP thus is found to be crucial for the formation of the prominent domain boundaries at both the *dif* and *oriC* regions, likely for promoting accessibility of these sites to proteins involved in the spatio-temporal regulations of DNA segregation and cell division[31,32].

**Transcription regulators induce secondary domain boundaries**. Although the average DNA density distribution clearly showed one peak at each of the two chromosomal arms (Fig. 3e), individual chromosomes typically contain a larger number (3–8) of physical domains (Fig. 2b). We examined the origin of the secondary domain boundaries by quantifying their distribution in the presence and absence of the nucleoid-associated proteins (NAPs) HU, H-NS, and Fis, which function as global transcription regulators[33]. Whereas all NAP-mutant cells preserved the overall M-shape DNA density distribution with a deep minimum at *dif* and a less pronounced minimum at *oriC* (Supplementary Fig. 14), deletion of NAPs could affect the domain boundaries: While deleting H-NS proteins had little effect, omitting HU and Fis proteins led to a very significant (near 80%) loss of the domain boundaries in the central regions of the two arms, which instead became significantly more enriched with domain centres (Fig. 3g, h, and see Supplementary Fig. 17 for a detailed analysis). Given that HU and Fis are transcription activators that localize

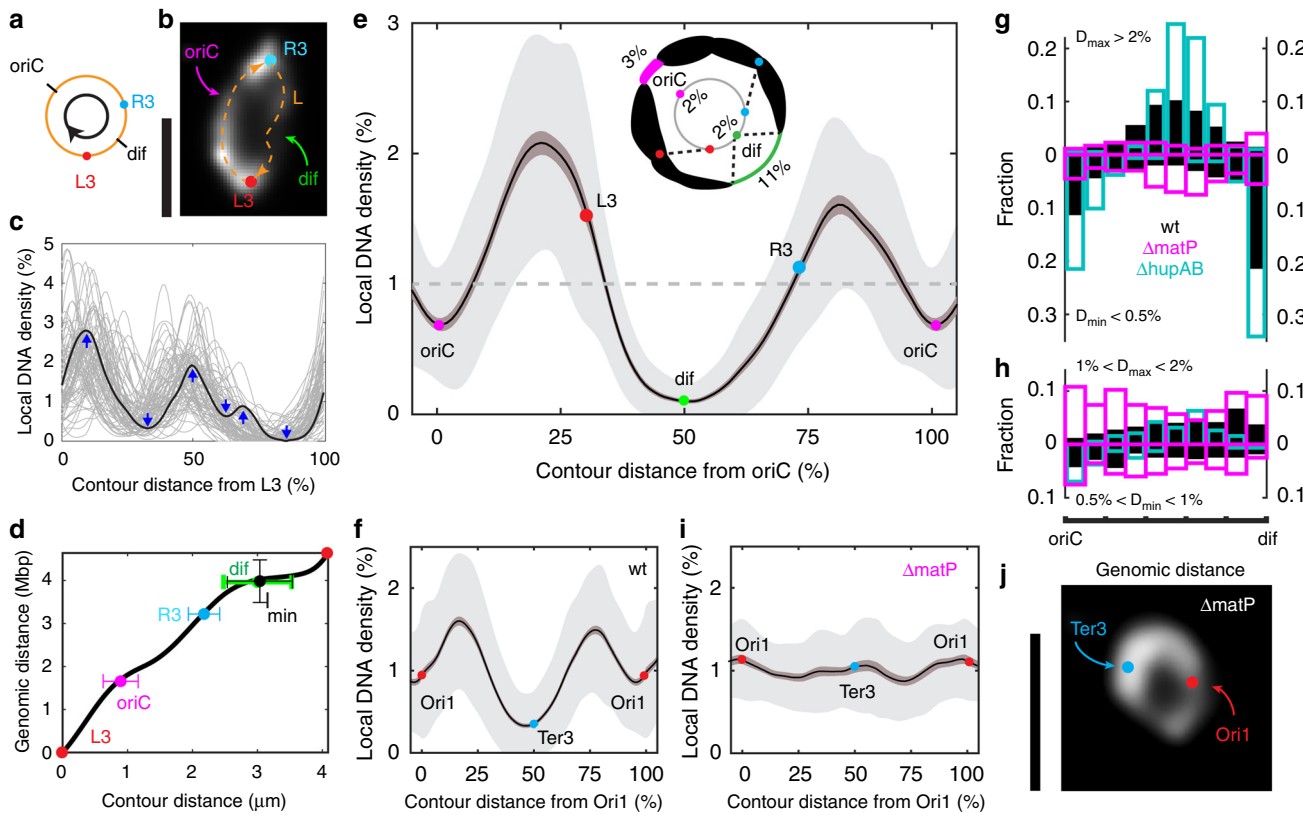

**Fig. 3** DNA density mapping along the circular *E. coli* genome. **a** Schematic of the *E. coli* genome with two FROS markers at the left and right arms shown in b-e. The arrow indicates the direction of density mapping. **b** Example of a circular chromosome as indicated in A, with the HU-mYPet fluorescence intensity shown in grey scale (deconvolved image). Scale bar, 2 μm. **c** Local DNA density along the ridge line of single circular chromosomes plotted as a function of percentile distance from L3. Each line indicates a single chromosome. An example chromosome density is highlighted in black. Blue arrows indicate local maxima (peaks) and local minima (valleys) in this example curve. $n = 82$. **d** Average cumulative density function mapping the genomic coordinates to the contour coordinates along the ridge of the circular chromosome. Marks indicate the measured positions of the R3 locus and global minimum $l_{min}$, and the predicted positions of *oriC* and *dif* sites. Error bars indicate s.d. **e** Local DNA density plotted versus genomic coordinate in percentile distance along the ridge line from the predicted *oriC* sites, with mean values and positions of *dif*, L3 and R3 indicated. Dark and light shading indicates s.e.m and s.d. Inset: schematic illustrating the DNA density distribution along a typical circular *E. coli* chromosome as concluded from the blob analysis and contour density analysis. $n = 292$. **f** DNA density distribution of a second independent strain with Ori1/Ter3 labels, plotted as in panel **e**. Note that in this strain left and right arms can be distinguished less well due to symmetry (Fig. 1a). $n = 74$. **g**, **h** Distributions of local maxima (plotted upwards) and local minima (downwards) in the DNA intensity within individual chromosomes along the genomes from *oriC* to *dif* site. Bin size 5% of the genome length (230kbp). Local maxima and minima are identified as in **c**. 'D' denotes local DNA density. **i** DNA density distribution of a *ΔmatP* strain with Ori1/Ter3 labels, plotted as in panel **e**. **j** Example of a circular chromosome in a *ΔmatP* cell, with the HU-mYPet fluorescence intensity shown in grey scale (deconvolved image). Scale bar, 2 μm.

sequence-nonspecifically throughout the genome (Supplementary Figs. 14 and 17), they likely do not produce domain boundaries directly, but instead promote their formation indirectly by stabilizing supercoils[34,35] within domains. These data, and our observations from antibiotic treatments (Supplementary Fig. 6), indicate that active transcription and the associated modulation of supercoil stability by HU, and Fis are essential for the emergence of the secondary domain boundaries across the genome. The strong cell-to-cell variations suggest that these domains are dynamic in nature.

***E. coli*'s chromosome is highly dynamic**. Indeed, very pronounced dynamics are apparent in time-lapse imaging of the donut-shape chromosomes. Figure 4a shows an example of a 2D-SIM movie that displays remarkable up-to-Mbp rearrangements in the chromosome morphology at a sub-minute timescale (see Supplementary Video 1). Our approach allowed us to construct coarse proximity maps of a single genome within a single live cell

over time (Fig. 4b, Supplementary Fig. 18, Supplementary Video 2), which score, at low resolution, the spatial proximity according to the physical positions of genomic loci along the toroidal chromosome. The prominent domain boundary at the *dif* region was very persistent, and the weaker one at *oriC* region was present as well in most of the frames. Domain boundaries outside the *oriC*/*dif* regions were observed to change in distinct steps between consecutive frames. These rough proximity maps are somewhat reminiscent of Hi-C maps that describe the contact frequencies between genomic loci in an ensemble of chemically fixed cells[8,9,18], but are also characteristically different as they measures the real-time toroidal distance within a single genome in a live cell.

The autocorrelation function of the density distributions along both the spatial (Fig. 4c) and genomic coordinates (Fig. 4d) decayed exponentially, with a decay half-time slightly smaller than 30 s, two orders of magnitude quicker than cell cycle time. Similar time constants for local chromosomal rearrangements were reported in rod-shaped cells[17], suggesting that the

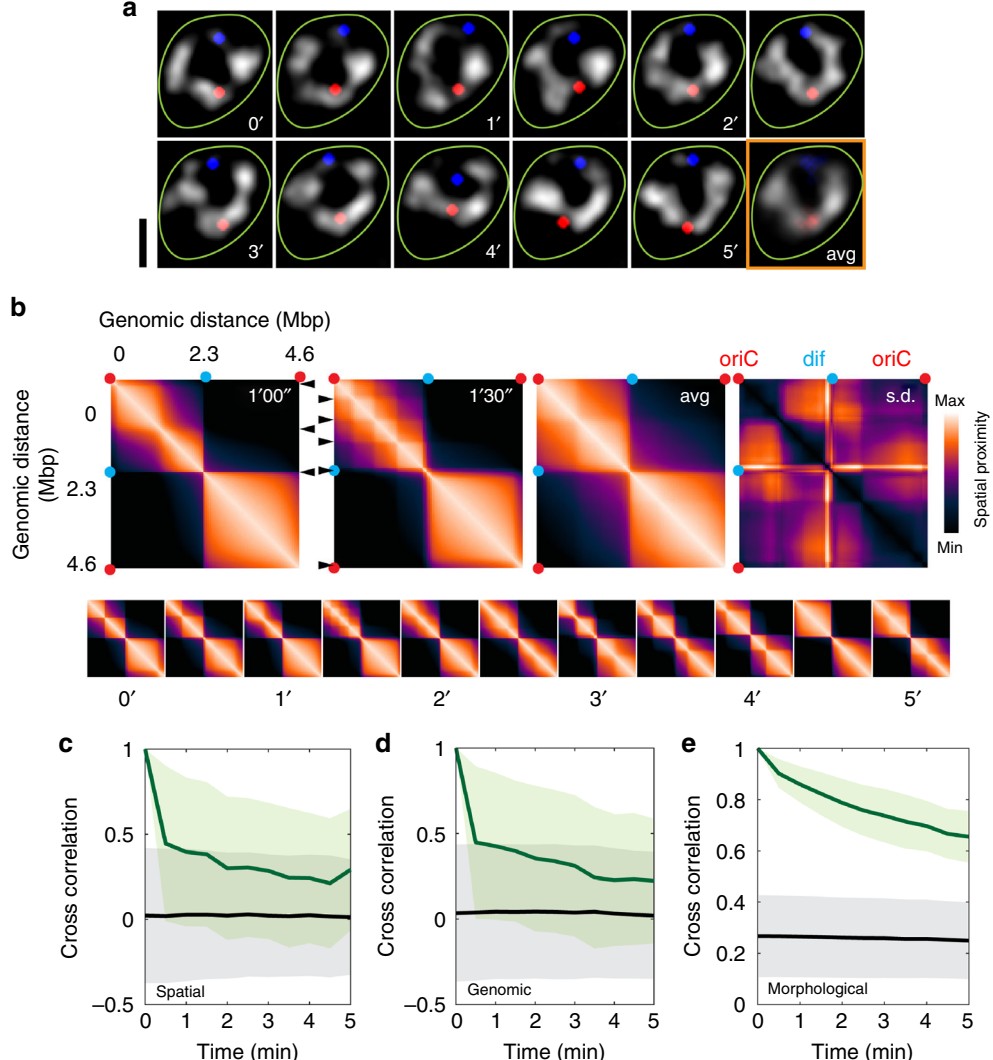

**Fig. 4** Temporal dynamics of the circular chromosome of *E. coli*. **a** Time-lapse structured-illumination microscopy (2D-SIM) images of a circular chromosome. Chromosomes in grey scale, Ori1/Ter3 localizations shown in red/blue, cell boundary in green. Time stamps are in minutes. The last frame is the time-average. Scale bar, 2 μm. **b** DNA spatial proximity maps derived from a single chromosome shown in a. Color bar indicates the level of spatial proximity for genomic loci along the circular chromosome. Arrowheads indicate apparent domain boundaries. Top panels, two consecutive frames, the time-average, and standard deviation values. Bottom panel, all 11 time frames. Red and cyan dots indicate Ori1/Ter3 loci. **c–e** Cross-correlation functions of individual chromosomes over time (green) compared to that across the cell population at each time point (black). These cross-correlation functions are measured with regard to the DNA density distribution along the contour in, respectively, spatial coordinates (**c**) genomic coordinates (**d**), and their morphology (**e**). Shown are mean and standard deviation values. $n = 46$

chromosome dynamics that we observe in these circularly shaped cell are depicting genuine chromosome rearrangements. Contrasting these fast density shifts, the full-image autocorrelation function of the chromosome morphology showed a much slower decay half-time, exceeding 5 min (Fig. 4e). We thus find that whereas the global configuration of the *E. coli* chromosome is quite stable within a given cell boundary, local DNA arrangements are plastic, which likely is important for fast regulations of gene expression.

## Discussion

The direct imaging of widened live *E. coli* cells presented here provides a unique window to understanding the bacterial genome organization. By decoupling DNA replication and cell growth in widened bacteria, we were able to visualize the circular shape of the chromosome in live *E. coli* cells. We found that the chromosome was organized in a torus topology with, on average, a lower density at the origin of replication and an ultrathin flexible string of DNA at the terminus of replication. The organization of the chromosome into a torus geometry is very different from the previously observed stiff arc organization in spheroplasts[11]. It is furthermore important to emphasize that our cell-widening method is a gentle method that is not dependent on the use of a temperature or an osmotic shock as previously used[11]. Using multiple controls (bacterial re-growth after treatment with A22 (Supplementary Fig. 1), various antibiotic treatment (Supplementary Fig. 6), and a shift to stationary phase (Supplementary Fig. 6D)), we concluded that cells are maintained in a normal physiologically active state.

The thin filamentous structure of the terminus region that we observe was proposed in earlier studies, based on the positioning of local foci relative to the cell shape in slowly growing AB1157 cells[2,15]. Contradicting results that were obtained in the MG1655 strain, however, claimed instead that the terminus was compacted[13] and suggested that an extended filament might have been specific to the AB1157 strain used in the earlier experiments.

Based on our direct imaging, however, we clearly observe, that the terminus is significantly extended into a filament in both strains (Figs. 1 and 3, Supplementary Fig. 7). Our observation suggests that the discrepancy between the previous observations may have been caused by differences in the cell cycle. Previous studies typically used replicating cells, which can lead to cells being in various growth phases with a varying number of chromosomes. For actively replicating cells, the flexible Ter region can be localized to the cell centre due to its interaction with the divisome components[32], which can lead to an appearance of a self-interacting domain. To resolve the finer details of chromosome organization, it is therefore important to have the cells synchronized and the chromosome number fixed—as our single-cell studies also show.

At the single-cell level, we found the chromosome to be strikingly heterogeneous and dynamic. Earlier super-resolution studies reported local high-density DNA regions (HDRs) in Bacillus subtilis and E. coli chromosomes[18,21], which show similarities but also some differences with the domains discussed here: While their genomic position was unresolved, the HDRs were observed to preferentially position at midcell for nascent nucleoids, or at 1/4 and 3/4 positions within the chromosome during the replication cycle[18,21]. Our domains are typically a bit less abundant per cell, with an average number of ~4 domains per single chromosome (Fig. 2b, Supplementary Fig. 11), compared to ~8 HDRs that were reported, depending on the cell growth media[18,21]. Furthermore it is interesting to note that the number of six domains was proposed by the macrodomain model[4]. Unlike previous studies on HDRs that only reported snapshot images, thus leaving the dynamic behavior unknown, our time-lapse study showed that domains undergo major dynamic rearrangements, splitting and merging and changing position at a minute timescales.

In contrast to the macrodomain model, we found the domain borders to be highly dynamic. The blob number was also influenced by the deletion of MatP proteins in cells, indicating that they regulate long-range interactions. Persistent domain boundaries at the origin and terminus of replication were found to be induced by MatP protein. Deletion of MatP resulted in a dramatic disappearance of the domain boundaries at Ori/Ter, presumably due to the interaction with the MukBEF SMC protein, which constitutes a major chromosome structuring protein in E. coli[30,31]. In a recent Hi-C study[8], Lioy et al. showed that MukBEF organizes the genome at large (hundreds of kbp) length scales and binds throughout the entire E. coli genome. MatP is known to bind only to the Ter region[13], and the authors proposed that it excludes MukBEF from binding, hence keeping the Ter inaccessible to MukBEF[30]. In another study based on positioning of fluorescently labelled MukBEF, it was observed that MukBEF loads at Ter region and by interacting with MatP is displaced from Ter towards Ori where it localizes preferentially. While our data clearly show that MatP is of central importance in organizing the chromosomal structure near Ter and Ori, the detailed nature of these interactions remains unclear and calls for further research.

We also tested the role of major NAPs in structuring the chromosome. While the deletion of H-NS had, surprisingly, no influence, which may possibly be due to the upregulation of StpA protein[36], we found that transcription regulators HU and Fis induce weak transient domain boundaries throughout the genome. This might be explained by the role both FIS[37] and HU[38] proteins play as regulators of supercoiling levels in E. coli. NAPs are also known to bind to specific DNA sequences[29,39] and locally introduce deformations in the DNA such as kinks, loops, and bridges[40]. Recent in vitro studies showed that HU and Fis proteins increase the dynamics as well as decrease the stiffness upon binding to supercoiled DNA[41]. Such additional mechanisms may also be at play when forming transient boundaries by NAPs.

Our quantitative live-cell imaging on donut-shape chromosomes thus provides a physical basis for understanding the plastic and dynamic architecture of bacterial genomes. We expect that its intricate domain features may guide future chromosome-organization studies in a broader spectrum of bacteria, whose diversity is increasingly appreciated.

## Methods

**Strain construction.** E. coli strains exhibiting DnaC temperature sensitivity[42], endogenous HU-mYPet label[43], single NAP deletions ($\Delta fis$, $\Delta hns$, $\Delta matP$)[42], and Ori1/Ter3 labeling[44] were described previously. For fluorescence labeling of H-NS and Fis, linear fragments of mYPet::aph frt amplified from plasmid pROD61[44] (a kind gift from the David Sherratt Lab), were transformed into strain W3110 to produce strain FW2561 (hns-mYPet:: aph frt) and strain FW2564 (fis-mYPet::aph frt), respectively, through λ/RED recombination[45]. These two constructs were then transduced into strain RRL189 through P1 transduction, and cured of the kanamycin resistance through pCP20. The construct dnaC2 ΔmdoB::aph frt from FW1957 was transduced into these resulting strains to produce the final strains FW2612 and FW2614.

To construct hupAB double mutant, strain RRL189 was sequentially transduced with the P1 lysates of JW3964(ΔhupA::aph frt), and JW0430(ΔhupB::aph frt) through P1 transduction and pCP20 curing of the kanamycin resistance, and then transduced with the lysate of FW1957 to finally produce strain FW2767.

For L3/R3 foci labels, strain RRL66 (AB1157, L3::lacOx240-hygR, R3::tetOx240-accC1 ΔgalK::tetR-mCerulean frt, ΔleuB::lacI-mCherry frt), a kind gift from Rodrigo Reyes-Lamothe, was transduced with hupA-mYPet::aph from FW1551, cured of kanamycin resistance using pCP20, and transduced with dnaC2 ΔmdoB::aph frt from FW1957 to result in strain FW2698. For Ori1/R3 foci labels, strain RRL150 was cured of kanamycin and chloramphenicol resistance through pCP20, and was sequentially transduced with R3::tetOx240::accC1 from RRL66 and ori1:: lacOx240:: cat frt from a derivative of RRL189, and dnaC2 ΔmdoB::aph frt from FW1957 to result in strain FW2721.

To generate AJ2830 and AJ2836 strains, MG1655 and AB1157 strains, respectively, were first transduced with hupA-mYPet::aph from FW1551, then cured of kanamycin resistance using pCP20, and further transduced with dnaC2 ΔmdoB::aph frt from FW1957.

For CRISPRi inhibition of replication BN2177 strain was transformed with pdCas9rna3 plasmid[46]. All strains used in this study are listed in Supplementary Table 1.

**Growth conditions.** For genetic engineering, E. coli cells were incubated in Lysogeny broth (LB) supplemented, when required, with 100 μg/ml ampicillin (Sigma–Aldrich), 50 μg/ml kanamycin (Sigma–Aldrich), or 34 μg/ml chloramphenicol (Sigma–Aldrich) for plasmid selection, and with 25 μg/ml kanamycin or 11 μg/ml chloramphenicol for selection of the genomic insertions of gene cassettes.

To obtain circular chromosomes, we grew cells in liquid M9 minimum medium (Fluka Analytical) supplemented with 2 mM MgSO₄, 0.1 mM CaCl₂, 0.4% glycerol (Sigma–Aldrich), and 0.01% protein hydrolysate amicase (PHA) (Fluka Analytical) overnight at 30 °C to reach late exponential phase. We then pipetted 1 μl culture onto a cover glass and immediately covered the cells[47] with a flat agarose pad, containing the above composition of M9 medium as well as 6% agarose and 4 μg/ml A22. The cover glass was then immediately placed onto a baseplate and sealed with parafilm to prevent evaporation. The baseplate was placed onto the microscope inside a 40 °C incubator for all cell growth and all imaging, unless noted differently. Circular chromosomes generally were imaged after 2.5–3 h.

For treatment of circular chromosomes with antibiotics, the cells were inoculated in liquid M9 medium described above at 40 °C with 4 μg/ml A22, and then placed under an agarose pad as described above with the addition of 100 μg/ml rifampicin or 10 μg/ml ciprofloxacin. The cells were incubated for 15 min before being imaged. Control samples did not have drugs added.

To reinitiate DNA replication, we grew the cells under agarose pad as described above for 3 h, then moved the baseplate to room temperature for 10 min before placing it back onto the microscope inside the 40 °C chamber to prevent further new replication initiation and for imaging.

To induce stationary phase physiology the cells were grown in liquid M9 medium as described above at 40 °C with 4 μg/ml A22, and then placed under an 6% agarose pad containing supernatant of stationary cell culture.

CRISPRi inhibition of replication initiation was done in BN2177 cells transformed with pdCas9rna3 plasmid. All steps were done in M9 medium supplemented with 0.2% glycerol, 2 mM MgSO₄, 0.1 mM CaCl₂, and 1% LB at 30 °C. Overnight culture was inoculated at 1/100 dilution in fresh media and grown for 2 h at 30 °C, then aTC (200 ng/ml) was added to the culture and incubated for 2 additional hours. Next, cells were deposited on an agarose pad containing M9 medium and a22 (4 μg/ml) and incubated at 30 °C for 3 h before image acquisition.

**Fluorescence imaging**. Wide-field Z scans were carried out using a Nikon Ti-E microscope with a 100X CFI Plan Apo Lambda Oil objective with an NA of 1.45. The microscope was enclosed by a custom-made chamber that was pre-heated overnight and kept at 40 °C. DAPI was excited by Nikon-Intensilight illumination lamp through a blue filter cube ($\lambda_{ex}/\lambda_{bs}/\lambda_{em}$ = 363–391/425/435–438 nm). mCerulean was excited by SpectraX LED (Lumencor) $\lambda_{ex}$ = 430–450 through a CFP filter cube ($\lambda_{ex}/\lambda_{bs}/\lambda_{em}$ = 426–446/455/460–500 nm). mYPet signal was excited by SpectraX LED $\lambda_{ex}$ = 510/25 nm through a triple bandpass filter $\lambda_{em}$ = 465/25–545/30–630/60 nm. mCherry signals was excited by SpectraX LED $\lambda_{ex}$ = 575/25 through the same triple bandpass filter. Fluorescent signals were captured by Andor Zyla USB3.0 CMOS Camera. In each channel, 19 slices were taken with a vertical step size of 227 nm (in total 4.3 µm). 2D and 3D Structured Illumination Microscopy imaging was carried out using a Nikon Ti-E microscope and a SIM module. A 100X CFI Apo Oil objective with an NA of 1.49 was used. Samples were illuminated with 515 nm laser line and a Nikon YFP SIM filter cube. mYPet, mCerulean, and mCherry signals of the same sample were also captured through wide-field imaging using a Nikon-Intensilight lamp. Filter cubes used for the wide-field imaging corresponding to the SIM images were CFP filters ($\lambda_{ex}/\lambda_{bs}/\lambda_{em}$ = 426–446/455/460–500 nm), YFP filters ($\lambda_{ex}$ / $\lambda_{bs}$ / $\lambda_{em}$ = 490–510/515/520–550 nm), and RFP filters ($\lambda_{ex}/\lambda_{bs}/\lambda_{em}$ = 540–580/585/592–668). For 3D-SIM imaging, 19 slices were taken with a vertical step size of 100 nm (in total 1.8 µm). SIM image reconstruction was done by using NIS-Elements (version 4.51) software. During image reconstruction, special care was taken to use the recommended parameters to avoid reconstruction artefacts. Furthermore, care was taken to check for photo-bleaching during image acquisition (which was negligible), to minimize drift during imaging, and to avoid artifactual signatures in the Fourier transforms of the reconstructed images[48] (Supplementary Fig. 19).

**Bacterial growth experiments**. *E. coli* cells were grown on a clear-bottom 96-well plate (Nunc) with a final volume of 150 µl of solution in each well. The plates were loaded into an Infinite 200Pro fluorescence plate reader (Tecan, Männedorf, Switzerland) and incubated at 30 °C in the presence of various concentrations of A22 drug (0.4–12 µg/ml). Agitated samples were shaken with orbital agitation (2.5 mm amplitude) for a period of ~5–15 h. Cell density was measured at 600 nm at 15 min intervals, measured in biological triplicates.

**Deconvolution**. Image stacks of 19 slices were deconvolved using the Huygens Professional deconvolution software (Scientific Volume Imaging, Hilversum, The Netherlands), using an iterative Classic Maximum Likelihood Estimate (CMLE) algorithm with a point spread function (PSF) experimentally measured using 200 nm multicolor Tetrabeads (Invitrogen). The PSF of the single-frame wide-field images has a FWHM of 350 nm horizontally and 800 nm vertically. Deconvolution to a great extent reduced the out-of-focus noise in the images, which also lead to an improvement in lateral resolution. A deconvolved 200 nm bead has FWHMs of 270 nm laterally and 580 nm vertically. Due to the large vertical FWHM, inherent to wide-field imaging (including deconvolution), we find that the fluorescent signal at the central frame, rather than an integrated signal of all z frames, provides the best estimation of the local DNA density.

**Automated cell identification**. Phase contrast images were fed into a customized Matlab program to produce masks of cell boundaries, which then were used to allocate chromosomes and foci in other fluorescence channels. A manual correction and rejection process was carried out as a final step of quality control, to correct or reject cells when neighboring cells were too close to allow the automated program to distinguish. Chromosome foci numbers were then counted[43] to ensure that selected cells have a single chromosome copy.

**Automated blob analysis**. The blob analysis used an approach where a deconvolved focal plane image was subject to step-by-step stripping of the subsequent brightest Gaussian spots based on our measured PSF described above, until the image became blank (see Supplementary Figs. 8 and 9). The centres of the identified spots were then placed back into the image, where their mutual distances were evaluated (Supplementary Fig. 8B). When the two spot centres were found to be located at a distance below our imaging resolution, they were assigned to the same blob (Supplementary Fig. 8C). The average diameters and intensities of each blob were then measured for statistical analyses.

**Automated density analysis**. The density profile of every chromosomes was automatically measured in the average focal plane of that particular chromosome. The centre of mass of the circular chromosome was used to as the origin to create an angular coordinate system that assigns an angular (α) and radial (r) coordinate to each pixel. The circular chromosome was then sectioned along the angular axis into 100 bins, each 3.6° (Supplementary Fig. 12A). The intensity maximum of each section was identified and mutually connected to constitute the ridge line of the filament. The points of this ridge line were then used to locate an improved centre of mass using just these points. Using the new centre, the ridge line was identified again. This process was iterated several times until the ridge line no longer was updated. Since the chromosome is not an isotropic torus, the identified points along the ridge line are not evenly spaced. In order to create evenly spaced

coordinates, the filament was resampled with even spacing along the ridge line. The total intensity of each section was then computed to represent the local intensity along the ridge line (where the ridge line was termed contour distance in the plots). To map the fluorescence signal at a particular position on the ridge line to the genomic sequence, we summed all intensity values and calculated the proportion of intensity that each section corresponded to (Supplementary Fig. 12B–D). This proportion was thus translated into DNA content in each section and was used to map spatial coordinates into genomic coordinates as shown in Fig. 3d and Supplementary Fig. 12D. In order to calculate the width of the chromosome, first the FWHM was measured along the 100 bins (each 3.6°) of the chromosome (as described above) and then the values were averaged to yield a single FWHM value for a single chromosome.

**Spatial proximity map construction**. A cumulative density function was constructed in a clockwise fashion along the ridge line of a single chromosome (Supplementary Fig. 18A–D, also see Fig. 3d), in which the contour distance between two genomic loci was estimated as indicated above. This spatial distance value was then plotted in the format of a color map on the pixels of the spatial proximity map (Supplementary Fig. 18E).

## Data availability
The data underlying Figs. 1, 2, 3e and 3f are provided as a Source Data file. Datasets that were acquired and analysed during the current study are available from the corresponding author upon request.

## Code availability
The analysis codes that were used in the current study are available from the corresponding author upon request.

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

## Acknowledgements

We thank Jeremie Capoulade, Erwin van Rijn, Jelle van der Does, Louis Kuijpers, My Nguyen, Margot Guurink, and Linda Chen (Huygens) for technical assistance, and David Sherratt, Rodrigo Reyes-Lamothe, and Jean-Luc Ferat for bacterial strains. This work was supported by ERC Advanced Grant SynDiv (No. 669598) and the Netherlands Organization of Scientific Research (NWO/OCW) as part of the Frontiers of Nanoscience Program. F.W. acknowledges support by Rubicon fellowship. A.J. acknowledges support by the Swiss National Science Foundation (Grants P2ELP2_168554 and P300P2_177768).

## Author contributions

F.W. and C.D. conceived and designed the project. F.W., A.J., and J.W. constructed the bacterial strains. F.W., A.J., J.W., and X.Z. did the microscopy experiments. A.J. performed the bacterial growth experiments. J.W.J.K. led the image analyses. J.W.J.K., F.W., and X.Z. wrote the data analysis programs. All authors wrote the paper. C.D. supervised the project.

## Additional information

**Competing interests:** The authors declare no competing interests.

