## [Peer Review File · Nature Communications]

Reviewer #2 (Remarks to the Author):

In this interesting manuscript, Wu et al. provide a fresh look at chromosome organisation in *E. coli*. They use an original approach to widen cells to be able to employ a series of super-resolution imaging approaches to study the well-defined *E. coli* chromosome. The direct observation of chromosome structure and dynamics turns out to be a powerful tool to clarify aspects of *E. coli* chromosome behaviour. For example, the toroidal shape and dynamic domain organisation are beautifully documented. These features, especially concerning domain formation, are likely to bear relevance to chromosome behaviour in all organisms. The manuscript is well written and the data presented largely support the authors conclusions. The authors should consider the following points before I can recommend publication in Nature Communications.

1. Introduction, line 46-47. The authors introduce their unique experimental system to widen the *E. coli* cell and block replication initiation. This forms the foundation for much of the experiments contained in this study. Two additional sentences could help the reader to understand how A22 makes *E. coli* lemon-shaped and how *dnaC2* makes replication stop at 40 degrees.

2. line 88, the bundle thickness is reported, defined as the width at half maximum of the peak intensity. In addition, it would be interesting to report on the intensity decay as a function of the distance from the peak towards the chromosome periphery. Does such an intensity decay function contain information as to the organisation of the DNA?

3. line 106, the DNA length contained in each cluster was analysed and the authors arrive at a scaling function reminiscent of a self-avoiding polymer. Can the authors explain what this means in biological terms and what implications this might have for DNA organisation inside the chromosome?

4. line 180, based on the reduced domain compartmentalization in NAP deficient cells, the authors suggest that active transcription and the associated impact on supercoil stability underlies domain formation. NAPs introduce kinks into the DNA path, similar to histones in eukaryotes. An alternative possibility therefore is that frequent DNA kinking facilitates domain formation. A stiffer DNA without NAPs might be less amenable to domain formation. That possibility could be discussed. This is also relevant for the discussion (line 260).

5. line 218, 'we concluded that cells are maintained in a normal physiologically active state.' If this is indeed the case, can cells be returned to growth after the treatment? This would provide the most

compelling evidence that the treatment was well tolerated. However, it is not necessarily a condition that cells survive treatment for the data to be informative.

6. line 245 ‘...MukBEF SMC protein which constitutes a major chromosome structuring protein in E.coli’. The MukBEF SMC complex is indeed a key regulator of chromosome architecture. If the authors had any data on the impact of the MukBEF complex on chromosome structure in their experimental system, that would be extremely interesting.

Reply to the reviewers (original comments in black font; our response in blue font)

Reply to reviewer 1

The ms of Wu et al describes the use of optical microscopies to visualize the E. coli chromosome in cells with a broadened shape after treatment with A22. They find that replication-arrested chromosomes in these conditions display toroidal shapes. They then go on to quantify the inhomogeneous DNA regions that they detect around these toroids. While the general approach is interesting, there are several important technical and interpretation issues that need to be considered to ensure the reliability of results as well as their biological relevance.

We thank the reviewer for the attentive reading of the manuscript and the thoughtful observations and comments from which we benefitted. Below we respond to the questions that were raised :

- The authors just cite papers where conventional microscopies were used to do whole-genome imaging. However, bacterial chromosomes have been visualised at super-resolution in live cells (Marbouty, Mol Cell 2015; Stracy, PNAS, 2015; Le Gall, Nat. Comm 2016), but these studies are not cited nor discussed. Importantly, these studies already described and quantified the existence of inhomogeneous regions in the E.coli and B. subtilis chromosomes which were defined as High-density DNA regions, or HDRs. The manuscript should cite those studies and refer to their 'blobs' using the existing nomenclature. Otherwise, they should justify why they use a different nomenclature.

Upon this suggestion of the reviewer, we have now expanded the list of references and incorporated most of these as well as other references into our manuscript. Furthermore, we have added a discussion about these previously reported HDRs and the similarities and differences in terms of size and dynamics of these compared to the clusters observed in our images. This has been incorporated in a significantly rephrased paragraph on page 8.

- Authors claim that use of A22 treatment keeps cells in a fully physiological state. They cite a reference where it was shown that: 1) nucleoids were not decondensed after use of A22; 2) chromosomes were still segregated in presence of A22. This does not mean that chromosomes are organized in the same way or that they have been segregated equally well. The fact that chromosomes segregate does not mean that there are no physiological effects or that chromosome organization was not affected. More controls should be presented.

Although we think that the observation that cells can re-initiate replication and division is a crucial control as it clearly means that the cells are alive, we now performed additional controls as requested by the referee – see the growth curves and phase contrast images that are now reported in Figure S1. Based on these data, one can clearly conclude that cells are alive during our experiments (Fig.S1.A,B), as evidenced for example by the recovery of growth in fresh media after exposure to A22 drug at 40°C (Fig. S1.C). Despite being initially circularized, these cells even recover the typical rod shapes (Fig.S1.D) after removal of A22 drug after 24 hours of growth on LB at 30°C.

- The authors provide in Fig S5 a method for the identification and quantification of high-density DNA regions. This method is based on the localization of an intensity maximum and the subtraction of the contribution of this maximum from the overall fluorescence image signal. This process is repeated

iteratively to localize many maxima. This sort of approach is valid when there are a small number of species (e.g. fluorescent molecules) that make up the fluorescence signal detected. However, it is completely unjustified when signals are coming from the sum of a large number of single emitters (thousands) making a continuum at the resolutions of optical microscopes. This method is the basis of much of the analysis provided in Figures 2, 3, 4. Therefore, this has to be solid if one is to believe the interpretation of results and the thrust of their conclusions.

We like to clarify this point by stating that the method presented in Fig.S7 (previously Fig.S5) does for sure not aim to resolve single emitters, but identifies much larger clusters that do not depend on the number of emitters bound to the chromosome (which are many thousands in our case). In other words, the aim of the method is not at all to reconstruct a collection of single-emitter Gaussian spots, but to provide an efficient means of describing each cluster by a low number of Gaussian components of varying amplitudes. Typically, we find that a few Gaussian spots per cluster suffice to describe their intensities, as shown in Fig.S8.

To clarify this better, we now added Fig.S8 where we discuss the process of reconstructing the clusters in more detail as well as show more controls describing the method. In addition, we added an introduction in the Methods section that describes our cluster analysis, explaining the aim of the Gaussian spot decomposition of the clusters. Finally, we now also note explicitly in the text that this method is not about reconstructing single emitters, but just providing an efficient means to extract the cluster sizes.

- The method used to transform fluorescence intensity to DNA content (in bp) has not been validated. It would be very important to be able to do this conversion accurately, as it is the basis of many of the conclusions in the paper.

We entirely agree with the referee about the importance of this conversion, and we are happy to clarify this point further: Care was taken to subtract any cytosolic background inside the cell outside the chromosome structure. This was typically done by subtracting a severely low-pass filtered version of the image. When this was done, any residual signal corresponded to fluorescence coming from the chromosome. Since, given the optical resolution limit, the labelling can be considered homogeneous over the chromosome, the local intensity then does correspond to the local chromosome density, while the total fluorescence count amounts, per definition, to the full genomic content of 4.6 Mbp.

This is described now in detail on page 13.

- In Fig. S7 the authors describe a method to determine genomic coordinate from images of the whole chromosome and the locations of two FROS tags. While we agree that the location of these tags in the whole DNA image are easy to extrapolate after chromatic corrections have been applied, the estimation of the intermediate genomic loci is not trivial. The authors base their method on the assumption that there is a linear correspondence between genomic coordinate and linear distance on the ridge. There is absolutely no evidence that this is the case. And in fact, it is most likely not the case as the chromosome behaves as a very flexible polymer and therefore its path will not follow continuously around the toroid. Again, many of the conclusions in Figs. 3-4 are based on this quantification, making interpretation of results difficult. The authors need to provide solid evidence for the reliability of their method or its limits.

We thank the reviewer for raising this point. Overall, one should consider the coordinate along the donut structure more as a guidance to the global genomic location than as a strict ruler. Very locally, there may be indeed some deviations from linearity, but on the coarser scale of our optical resolution, the flexible polymer structure averages out and linearity is recovered. In fact, the precision of pinpointing the genomic position in the chromosome is in practice limited by the cell-to-cell variation and dynamics of moving clusters. This puts a spread of maximum $\pm 15\%$ on the expected position of a label (as shown in the inset of Fig.S10D, based on the ter marker).

Most importantly, the conclusions presented in Fig. 3 and Fig. 4 are not affected. For the density curves presented in Fig. 3, the cell-to-cell variations and dynamic motion have no significant influence on the data (as shown for the case with L3-R3 as well as ori1-ter3 foci), since the average density plots shows the same two pronounced gaps near ori and ter. Regarding the spread of the ter region, which is based on the local ter3 foci positioning, the pinning provides a very accurate estimate, as it can be seen from the sharp alignment of this ter minimum shown in Fig. S11F.

- Controls that 3D-SIM was appropriately used should be provided (e.g. use SIM-check).

We performed such SIM checks using the Nikon NIS-Elements software. We updated the text and now mention the reconstruction parameters, see the modified text on page 12.

- Images in Fig S5 seem to have been analysed in 2D (but are really 3D right). Are we missing something here?

Indeed the analysis is based on 2D focal plane images and not on a 3D volume reconstruction. Since, as mentioned in the paper, the chromosome is vertically confined to a $\sim 1\mu\text{m}$ height by the agarose pad and since the thickness of the chromosomes is $< 400\text{nm}$, it is well justified to take the focal plane of the chromosome as representative of the chromosome density.

- The authors make a claim that 'the torus topology is maintained by active physiological processes'. To support this they should provide experiments where energy sources are depleted.

These relevant controls with depletion of energy sources have been performed and the results of these experiments are provided in Fig.S5. Fig.S5B-C shows that the chromosome loses its donut topology upon blocking transcription by rifampicin or upon blocking the gyrase and topoIV enzyme activity by ciprofloxacin. Furthermore, Fig.S5D shows that the chromosome gradually condenses into a blob in the center of the cells upon a shift to stationary phase where the available energy sources run out. These complementary controls clearly confirm that the donut topology is maintained by active physiological processes.

- The authors used 3D-SIM in the first and last figures, but they do not seem to have used it in the others. It is not clear why, and whether the results would be the same. For instance, it seems that estimation of the thickness of the torus was obtained from epi-fluorescence imaging. As the resolution of this technique is limited by diffraction, what they measure is likely an overestimation of the thickness. What result would they obtain using 3D-SIM?

The vast majority of images and data analysis in our paper report 2D-SIM data. In the one exception of Fig.1D, we additionally report a 3D SIM image to clarify the chromosome topology in 3D. Furthermore, we now show more explicitly that the deconvolution results and SIM results are basically the same, for which we have added Fig.S4 to show time-average values of the chromosome width and contour length measured by 2D-SIM images. We updated the main text to refer to these new data on page 3.

- The authors make the claim that the physical characterization of the toroid would represent the basis for future polymer modeling to understand the role of confinement and volume exclusion in DNA segregation. However, these results are obtained from a somewhat abnormal system where chromosomes are decondensed due to the abnormal size of cells and due to the blockage in DNA replication. To make the claim that their results are relevant to bacteria under normal physiological conditions, authors should repeat the experiments in these conditions.

On this point, we disagree with the reviewer. From the multiple extensive controls that we did regarding cell growth and response to drugs and to changes in the growth media (Fig.S1 & S5), we can unambiguously conclude that the cells are metabolically active and in a relevant physiological state. In

our opinion, the observation of the toroidal, dynamic, and clustered structure of the chromosomal provides striking and entirely nontrivial insight into the fundamental organization of the bacterial chromosome that is of great interest and will provide a stimulus for future modeling of chromosomes in the confined volume of the cell.

- From analysis of chromosome ridges (see above), the authors conclude that there are regions of high and low DNA density located at Left/Righth or at oriC/Ter, respectively. They then go on to call the former 'domains' and the latter 'domain boundaries'. These terms have very specific connotations in chromosome architecture and regularly include the existence of factors that nucleate the organization of chromatin in domains or/and the delimits interactions between domains (typically by highly transcribed genes or by chromatin insulators). The authors should refrain from using these specific terms in the absence of a demonstration that these mechanistic implications exist for their system.

We realize that the hierarchical organizational structure of the chromosome involves domains at various length scales. We note that the language of domains is used not only to denote TAD domains separated by chromatin insulators and the like, but also was used before at the Mbp level in the macrodomain studies by Espeli et al (2008). Following the latter authors, we feel that it is very relevant to use the terms 'domains' and 'domain boundaries' for these Mbp clusters, since these structures naturally present themselves to us from the data, see e.g. the discussion on the data of Fig. 3E-I.

To accommodate the comment of the referee, we now added a brief discussion to page 5, to clarify our terminology.

- By analysing the distances between blobs the authors claim to produce maps 'reminiscent of Hi-C maps that describe the contact frequencies between genomic loci'. Their method does not provide a measure of contact frequency/probability. But most importantly, it lacks from genomic specificity (as they can only position precisely two genomic loci). Therefore the comparison is disingenuous.

The referee is correct that our method does not probe contact frequencies (and neither did we claim that). Our method *does* probe density correlations along the genomic coordinate, albeit with low resolution (very much lower than for the well-developed Hi-C methods). This is the reason that we mentioned a similarity to the chromosome capture methods. Indeed, high-chromosome-density regions will lead to higher contact frequencies in Hi-C data. Although we feel that this analogy is of some interest to note for the reader, we are not adamant on it (it is merely an analogy after all), and we could remove the statement if the editor prefers.

- The authors do an analysis of dynamics in Fig.4 and recover a 'local' time constant of ~ 30 s. Similar estimations have already been done in the literature and are not cited either here (see Yazdi, 2012).

We thank the referee for pointing out this reference. We added the citation to the manuscript and updated the text on page 6.

- The quantification in the number of high density regions (Fig 2B) should be compared to literature values.

We now added a paragraph in the discussion that compares HDRs and the clusters described in Fig.2B, see page 8

- Typically, treatment of cells with rifampicin decompacts the nucleoid, but this is not observed here. Why?

We agree with the referee that treating the cells with rifampicin for a long time (a few hours) typically leads to chromosome decondensation, as has been reported in the literature and which also is seen in

our experiments on lemon-shaped cells. In the experiments reported in the paper, however, we imaged cells very quickly (minutes) after the exposure to rifampicin, where we observe a loss of the donut topology and a collapse of the chromosome, which can be attributed to transcription blockage.

We now updated the text, highlighting that the antibiotic treatments were done only for short periods of time, and noting the similar decondensation on long exposures, as in other reports, see page 3.

Reply to Reviewer 2

In this interesting manuscript, Wu et al. provide a fresh look at chromosome organisation in *E. coli*. They use an original approach to widen cells to be able to employ a series of super-resolution imaging approaches to study the well-defined *E. coli* chromosome. The direct observation of chromosome structure and dynamics turns out to be a powerful tool to clarify aspects of *E. coli* chromosome behaviour. For example, the toroidal shape and dynamic domain organisation are beautifully documented. These features, especially concerning domain formation, are likely to bear relevance to chromosome behaviour in all organisms. The manuscript is well written and the data presented largely support the authors conclusions. The authors should consider the following points before I can recommend publication in Nature Communications.

We would like to thank the reviewer for the attentive reading of the manuscript, the thoughtful observations and comments, and the positive remarks about the originality, interest, and quality of our work. Below we respond to the questions that were raised :

- Introduction, line 46-47. The authors introduce their unique experimental system to widen the *E. coli* cell and block replication initiation. This forms the foundation for much of the experiments contained in this study. Two additional sentences could help the reader to understand how A22 makes *E. coli* lemon-shaped and how dnaC2 makes replication stop at 40 degrees.

We thank the reviewer for the suggestion and now expanded the introduction part to clarifying how A22 and temperature sensitivity functions, see page 2.

- line 88, the bundle thickness is reported, defined as the width at half maximum of the peak intensity. In addition, it would be interesting to report on the intensity decay as a function of the distance from the peak towards the chromosome periphery. Does such a intensity decay function contain information as to the organisation of the DNA?

We thank the referee for raising this interesting point. In the case of contact frequencies in HiC maps, the decay rate can give a good estimate of an event rate that is gradually decreasing. However, the chromosome width measurements are based on microscopy and the optical resolution is limiting in this case. We measured the chromosome width and contour length by SIM and added the figures in supplementary Fig.S4. From the data it is apparent that the distribution of the width is quite narrow, indicating that the chromosome width is well defined and the chromosome density does not decay very gradually. Hence, using the full width at half maximum of the peak appears to be the relevant measure.

- line 106, the DNA length contained in each cluster was analysed and the authors arrive at a scaling function reminiscent of a self-avoiding polymer. Can the authors explain what this means in biological terms and what implications this might have for DNA organisation inside the chromosome?

This is a good question. It is however rather hard to draw a firm conclusion based on this scaling alone. The observed scaling that is similar to that of a self-avoiding polymer may perhaps suggest that the chromosome consists of domains that interact only within domains and very weakly with each other. However, we prefer to refrain from such speculation and would rather like to wait for future polymer simulation studies on this aspect.

- line 180, based on the reduced domain compartmentalization in NAP deficient cells, the authors suggest that active transcription and the associated impact on supercoil stability underlies domain formation. NAPs introduce kinks into the DNA path, similar to histones in eukaryotes. An alternative possibility therefore is that frequent DNA kinking facilitates domain formation. A stiffer DNA without

NAPs might be less amenable to domain formation. That possibility could be discussed. This is also relevant for the discussion (line 260).

We do agree with the referee: Although both FIS and HU can induce a change in the distribution of gaps due to their activity as supercoiling homeostasis regulators, it is possible that there also are additional mechanisms at play. To address this point we added some text to page 8.

- line 218, 'we concluded that cells are maintained in a normal physiologically active state.' If this is indeed the case, can cells be returned to growth after the treatment? This would provide the most compelling evidence that the treatment was well tolerated. However, it is not necessarily a condition that cells survive treatment for the data to be informative.

It is indeed the case that the cells return to growth after the treatment, and indeed this is compelling evidence that the treatment was well tolerated. We now report these experiments in an additional supplementary Fig.S1. Despite being initially circularized, the lemon-shape cells even recover the typical rod shapes after removal of the A22 drug.

- line 245 '...MukBEF SMC protein which constitutes a major chromosome structuring protein in E.coli'. The MukBEF SMC complex is indeed a key regulator of chromosome architecture. If the authors had any data on the impact of the MukBEF complex on chromosome structure in their experimental system, that would be extremely interesting.

We do agree with the referee that understanding the role of MukBEF SMC complex in organizing the chromosome is a very interesting topic. However, such a study is technically very challenging and beyond the scope of this manuscript. We are currently making first steps to pursue this as a follow up project.

Reviewers' comments:

Reviewer #1 (Remarks to the Author):

1. Instead of discussing the literature on DNA imaging using live super-resolution microscopies in the introduction, the authors first mention them in the Discussion. This literature dates back from 2015 and 2016, it is thus not clear why they would not be cited in the introduction.

1.a In the discussion of HDRs, the authors make a number of misrepresentations. For instance, it is claimed that HDRs are static and position at $\frac{1}{4}$ or $\frac{1}{2}$ of cell lengths. In the original papers, it was shown that the maxima of the probability of finding HDRs were at these positions, not that they were statically located there. This has to be corrected in the Discussion.

1.b The domains observed in the current manuscript display movements in the minutes time-scales, and this is used as an argument to discriminate them from HDR. But HDRs were mostly measured in snapshots, so no comparison can be made related to dynamics between these two structural features. This needs to be changed in the Discussion.

1.c The authors cite a mean number of HDRs for vegetatively growing cells where 1.5-2 chromosomes are present. In the original studies, it was shown that cells with a single chromosome (what is being used in the current study) have in average 4-10 HDRs, very comparable to the 3-8 domains observed for a single chromosome in this study. Thus, in our opinion, this cannot be used either to discriminate between these structural features.

1.d Lastly, the authors claim that the average size of HDRs was reported to be 230kb. The Discussion now claims that the domains visualized in this manuscript are much larger "up to Mbp-size". But the distribution shown in Fig. 2D peaks at ~300kb. Thus, we are not convinced this is a strong argument either.

In short, we are not convinced by their arguments about the differences between HDRs and the domains detected here.

2. A second issue in the original review was with the method for quantification of clusters. The authors now provide further explanation and simulations in Supp. Fig. 8.

The mention in the original review of single emitters was to explain when such an approach would be valid, it was not because this reviewer thought the authors were trying to quantify single emitters. In the simulations in Supp. Fig 8 the authors seem to show that they are able to recover "perfectly" ground truth structures from their method. But they fail to show error bars, which would indicate how often the method fails to detect properly ground-truths, and they consistently tend to simulate symmetric situations. To really validate the method, the positions of the simulated clusters should be randomized (as chromosome organization is known to be highly heterogeneous/dynamic) to more faithfully reproduce experimental conditions. Also, simulations do not seem to have considered background or noise. These will likely affect their results.

Overall, we are not convinced by their arguments that their method for detection of clusters is valid. This would call into question what they are really detecting by this analysis method.

3. In the original, review we stated: "...the method used to transform fluorescence intensity to DNA content (in bp) has not been validated".

The authors further explain the method but do not seem to provide validation. Such a validation could potentially be obtained from their experiment where they detect DNA volume and the positions of FROS. In this case, they know the genomic position of each FROS and they can measure the total integrated DNA intensity between two given FROS. Then they can validate whether their measurement based on DNA intensity is consistent with the genomic distance between FROS probes.

4. Originally, we asked for controls that 3DSIM was appropriately used (e.g. using SIMcheck). The authors do not provide such controls, and just mention the reconstruction parameters, which is not what was requested.

5. In the original review, we asked why measurements were based on 2D imaging. The authors do not provide evidence using 3D and 2D imaging to reassure this reviewer. Rather they just argue that their method is correct provided the axial thickness of the chromosome...

In similar imaging conditions, it is known that FROS often come in and out of the imaging plane if one does 2D imaging. Therefore, a considerable contribution to the fluorescence signal is likely neglected by performing 2D imaging.

Even more troubling is the new information that most of the imaging in the paper is done using 2D-SIM. If this is the case, it seems that the arguments of the authors regarding the depth of field of widefield imaging being enough to capture the whole chromosome may be invalid.

Thus, I am still not convinced about the validity of 2D imaging for the type of analysis they do in this manuscript. It would be desirable to see experimental evidence for their claims.

6. Differences between SIM and widefield. The authors now claim that they quantify bundles widths and lengths by SIM and widefield and obtain the same results. They then argue that it is equivalent to use either method. Surely, there is something we don't understand, because bundle widths are ~450nm (mean) by widefield (Fig. 1I) and ~300 nm by 2D-SIM. Are we missing something?

As a note, in the text it is indicated that the mean bundle width is 0.4 μm . But, the distribution shown in Fig. 1I points to a very different value of at least 450nm or higher.

7. Originally we remarked that the experimental conditions used in this manuscript (where cells are lemon-shaped) are not typical of bacterial shapes. Thus, a claim that these results can be used for future modeling of chromosome structure is far fetched. We understand the arguments of the authors, but we continue to think that chromosomes under normal circumstances (i.e. not in lemon shaped cells) may behave differently than in normal growth conditions where they adopt their physiological shapes.

Reviewer #2 (Remarks to the Author):

I read the revised manuscript "Direct imaging of the circular chromosome in a live bacterium" by Dekker and co-workers. The authors have satisfactorily addressed my concerns and I can now recommend this manuscript for publication

Reply to reviewer 1 (original comments in black font; our response in blue font)

We thank the reviewer for the thorough reading of the manuscript and the detailed observations and comments. Below we respond to the questions that were raised :

1. Instead of discussing the literature on DNA imaging using live super-resolution microscopies in the introduction, the authors first mention them in the Discussion. This literature dates back from 2015 and 2016, it is thus not clear why they would not be cited in the introduction.

We adopted the referee's suggestion and moved the super-resolution literature citations on live cells to the introduction.

1.a In the discussion of HDRs, the authors make a number of misrepresentations. For instance, it is claimed that HDRs are static and position at $\frac{1}{4}$ or $\frac{1}{2}$ of cell lengths. In the original papers, it was shown that the maxima of the probability of finding HDRs were at these positions, not that they were statically located there. This has to be corrected in the Discussion.

The referee is correct that the maxima of the probability of finding HDRs were at these positions but that the HDRs were not necessarily static. We updated the text and rephrased the statements, as can be seen on page 8.

1.b The domains observed in the current manuscript display movements in the minutes time-scales, and this is used as an argument to discriminate them from HDR. But HDRs were mostly measured in snapshots, so no comparison can be made related to dynamics between these two structural features. This needs to be changed in the Discussion.

We agree that information on the dynamics of HDRs was not reported in the previous papers, so no direct comparison can be made regarding this particular point. We updated the text and rephrased the statements, as can be seen on page 8.

1.c The authors cite a mean number of HDRs for vegetatively growing cells where 1.5-2 chromosomes are present. In the original studies, it was shown that cells with a single chromosome (what is being used in the current study) have in average 4-10 HDRs, very comparable to the 3-8 domains observed for a single chromosome in this study. Thus, in our opinion, this cannot be used either to discriminate between these structural features.

We agree that the range is similar, though the average numbers are different (4.5 ± 1.1 in our cells versus 7.9 ± 2.1 reported for the HDR paper of Marbouty et al, 2015). We have updated the text and rephrased the statements, as can be seen on page 8.

1.d Lastly, the authors claim that the average size of HDRs was reported to be 230kb. The Discussion now claims that the domains visualized in this manuscript are much larger "up to Mbp-size". But the distribution shown in Fig. 2D peaks at ~ 300 kb. Thus, we are not convinced this is a strong argument either.

We made our statement on the domain size more precise and softened the claims regarding the differences between the domains and HDRs, see page 8.

In short, we are not convinced by their arguments about the differences between HDRs and the domains detected here.

We note that the similarities between the domains that we report and the HDRs that were reported before are of some interest but not the crucial element of our paper. The referee is correct that there are similarities and we now acknowledge these clearly and we refer to the work published before. Furthermore, we note some differences between them in terms of their definition, average number, and size.

2. A second issue in the original review was with the method for quantification of clusters. The authors now provide further explanation and simulations in Supp. Fig. 8.

The mention in the original review of single emitters was to explain when such an approach would be valid, it was not because this reviewer thought the authors were trying to quantify single emitters. In the simulations in Supp. Fig 8 the authors seem to show that they are able to recover "perfectly" ground truth structures from their method. But they fail to show error bars, which would indicate how often the method fails to detect properly ground-truths, and they consistently tend to simulate symmetric situations. To really validate the method, the positions of the simulated clusters should be randomized (as chromosome organization is known to be highly heterogeneous/dynamic) to more faithfully reproduce experimental conditions. Also, simulations do not seem to have considered background or noise. These will likely affect their results.

Overall, we are not convinced by their arguments that their method for detection of clusters is valid. This would call into question what they are really detecting by this analysis method.

We agree with the referee's point that the positions of the simulated clusters can be randomized to further validate our method. Hence, we now expanded our simulations to the case of randomly distributed clusters, including repeats per setting to obtain error bars on the results. An excellent tracing of individual clusters is observed for cluster numbers of 3 to 5, while small deviations (of max 0.5 cluster count) are seen at lower and higher numbers (Fig.S9C-E).

Furthermore, we now expanded our simulations to evaluate the effects of random noise and deconvolution, and we quantified the noise in our experiments to compare it with our simulations. Gratifyingly, we find that only minor deviations (counting error ~ 0.2 counts) are obtained in the average number of clusters for realistic noise levels (Fig.S9FG).

We conclude that our cluster algorithm is robust and sensitive enough to observe subtle shifts in cluster distributions for realistic deconvolved images and for the relevant levels of signals and noise intensities. We expanded the description of the simulations in SI Fig.S9 and added the extensive simulations data for random and noisy clusters.

3. In the original, review we stated: "...the method used to transform fluorescence intensity to DNA content (in bp) has not been validated".

The authors further explain the method but do not seem to provide validation. Such a validation could potentially be obtained from their experiment where they detect DNA volume and the positions of FROS. In this case, they know the genomic position of each FROS and they can measure the total integrated DNA intensity between two given FROS. Then they can validate whether their measurement based on DNA intensity is consistent with the genomic distance between FROS probes.

We agree: To validate our results, we used three different strains with different sets of FROS tags (ori1-ter3 and left3-right3 as previously shown in Fig.3E and F in the main text, and we now also added new experimental data for FROS pairs with ori1-right3). While the ori1-ter3 foci do segment the chromosome into almost symmetric halves (51:49%) the other two tags, left3-right3 (70:30%) and the ori1-right3 (34:66%) segment the chromosomes into clearly asymmetric halves. This asymmetry is much larger than our measurement error, which allows us to unambiguously distinguish the two arms based on the DNA intensity difference, as discussed in the main text on page 3 and 4. Our protocol for the deduction of the genomic position as function of distance was previously already explained in Fig.S12D, but we expanded our explanation now substantially.

The fact that these three strains with different FROS variants lead to essentially the same result validates the method. We now added a more detailed explanation and description of controls of how we transform fluorescence intensity to DNA content, see the supplementary Fig. S11.

4. Originally, we asked for controls that 3DSIM was appropriately used (e.g. using SIMcheck). The authors do not provide such controls, and just mention the reconstruction parameters, which is not what was requested.

We thank the referee for clarifying what he/she asked for. Following the suggestion of the referee, we now performed SIM imaging controls with SIMcheck software, showing that imaging was performed without imaging and reconstruction artifacts. We added these controls in supplementary material now, see Fig.S19.

5. In the original review, we asked why measurements were based on 2D imaging. The authors do not provide evidence using 3D and 2D imaging to reassure this reviewer. Rather they just argue that their method is correct provided the axial thickness of the chromosome...

In similar imaging conditions, it is known that FROS often come in and out of the imaging plane if one does 2D imaging. Therefore, a considerable contribution to the fluorescence signal is likely neglected by performing 2D imaging.

Even more troubling is the new information that most of the imaging in the paper is done using 2D-SIM. If this is the case, it seems that the arguments of the authors regarding the depth of field of widefield imaging being enough to capture the whole chromosome may be invalid.

Thus, I am still not convinced about the validity of 2D imaging for the type of analysis they do in this manuscript. It would be desirable to see experimental evidence for their claims.

We would like to clarify that most of the imaging reported in the paper is not done using SIM imaging but based on widefield imaging with subsequent deconvolution. Apparently, this was unclear, so to avoid any further confusion to readers, we now added a brief description to every image in the main text or caption, to indicate whether it is a SIM or wide-field deconvolved image.

SIM imaging has its drawbacks - long exposure times and high laser excitation, which makes it challenging for 3D dynamic imaging of live bacterial samples. In our experiments, we thus compared the deconvolution of widefield 3D-Z stacks with SIM images to confirm our quantitative findings. We used the complementary features of both approaches to achieve high spatial or temporal resolution in live cells without cell fixation. Of course, we look forward to seeing future improvement in imaging techniques that can improve spatial and temporal resolution simultaneously.

In our report, we indeed assumed that the 3D structure of the chromosome can be sufficiently approximated by its 2D projection to the focal plane. We agree that this should be verified, since in doing so, one could, for example, observe apparent low-density areas along the circular donut structure where in reality the structure just extends out of the plane. We now added a new Fig.S4 to show why the approximation is valid in our case. Indeed, there is some bending of the chromosome into the Z-planes but it is typically within ± 1 Z plane (227nm) from the focal plane, and the typical error in measuring DNA density that can arise due to the off-plane signal is of order of 5-10% of the average intensity.

Finally, we note that the fact that chromosome lies rather flat in one plane actually highlights an advantage of our approach where we maintain the height of the cells close the wild-type cell diameter.

6. Differences between SIM and widefield. The authors now claim that they quantify bundles widths and lengths by SIM and widefield and obtain the same results. They then argue that it is equivalent to use either method. Surely, there is something we don't understand, because bundle widths are ~ 450 nm (mean) by widefield (Fig. 1I) and ~ 300 nm by 2D-SIM. Are we missing something?

As a note, in the text it is indicated that the mean bundle width is $0.4 \mu\text{m}$. But, the distribution shown in Fig. 1I points to a very different value of at least 450 nm or higher.

We would like to thank the reviewer for pointing out this discrepancy. Upon this comment, we thoroughly re-evaluated the data with respect to the FWHM of the cross-sectional profiles of the chromosomes, upon which we realized that the width measured by SIM imaging was indeed underestimated due to the Gaussian filtering step during the image analysis. Hence, we re-measured the width of the chromosomes on the SIM images and now obtain a good agreement between the deconvolved and SIM values, with numbers for the FWHM of $0.45 \pm 0.05 \mu\text{m}$ and $0.40 \pm 0.02 \mu\text{m}$, respectively, and a contour length of $4.0 \pm 0.6 \mu\text{m}$ and $4.0 \pm 0.8 \mu\text{m}$, respectively. We thank the referee for catching this glitch, and we updated the Fig.S5 to incorporate the updated SIM dataset.

Regarding the exact value of the mean bundle width, we corrected the typing mistake (page 3).

7. Originally we remarked that the experimental conditions used in this manuscript (where cells are lemon-shaped) are not typical of bacterial shapes. Thus, a claim that these results can be used for future modeling of chromosome structure are far-fetched. We understand the arguments of the authors, but we continue to think that chromosomes under normal circumstances (i.e. not in lemon shaped cells) may behave differently than in normal growth conditions where they adopt their physiological shapes.

At this point we respectfully disagree with the referee. The controls show that the cells are maintained in a physiologically active state and the data reveal relevant, new, and interesting information on the genomic organization. Obviously, confinement itself has its own merits to constrain the chromosome, but in our opinion, the current new data provide very interesting new insights into the properties of the bacterial chromosome.

REVIEWERS' COMMENTS:

Reviewer #1 (Remarks to the Author):

The authors have now answered satisfactorily all our concerns. In a couple of points, like the relevance of lemon-shaped cells, we agree to disagree.